# Does Overqualification Play a Promoting or a Hindering Role? The Impact of Public Employees’ Perceived Overqualification on Workplace Behaviors

**DOI:** 10.3390/bs14010048

**Published:** 2024-01-12

**Authors:** Zhe Shang, Chenhui Zuo, Yan Shi, Ting Zhou

**Affiliations:** 1School of Government, Beijing Normal University, Beijing 100875, China; zshang@bnu.edu.cn (Z.S.); zuochenhui@mail.bnu.edu.cn (C.Z.); bjsfdx_202306@163.com (Y.S.); 2Department of Medical Psychology, School of Health Humanities, Peking University, Beijing 100191, China

**Keywords:** perceived overqualification, organizational citizenship behavior, workplace deviance behavior, psychological entitlement, perceived control

## Abstract

Drawing upon the conservation of resource theory, we offer a framework for understanding the mechanism underlying the effect of public employees’ overqualification on their cognitive and behavioral outcomes, through both positive and negative paths. We propose that perceived overqualification elicits two subjective cognitions, namely, perceived control and psychological entitlement, which further lead to public employees’ behaviors through approach (organizational citizenship behavior) and avoidance (workplace deviance behavior) tendencies. A total of 421 public employees participated in the three-stage time-lag investigation. The findings indicated that public employees’ perceived overqualification is positively related to perceived control, and perceived control is positively related to organizational citizenship behavior. Perceived control completely mediates the relationship between perceived overqualification and organizational citizenship behavior. Perceived overqualification is positively related to psychological entitlement, and psychological entitlement is positively related to workplace deviance behavior. Psychological entitlement completely mediates the relationship between perceived overqualification and workplace deviance behavior. This study constructed a double-edged sword model of perceived overqualification based on the public sector, discussing the influence of public employees’ perceived overqualification on their workplace behaviors from the perspective of resource assessment and self-evaluation, and providing theoretical guidance for the practice of human resource management within the public sector.

## 1. Introduction

When people believe that their abilities or experiences are much more than the job criteria, they may feel like “big fish in a small pond”, which is known as perceived overqualification (POQ) [1,2,3]. In recent years, with the slowdown of economic development and the expansion of higher education in universities, the competition in the labor market has intensified, and POQ has become a familiar and important issue, receiving increasing attention [4,5]. This phenomenon is not only prominent in the private sector, but also gradually becomes a prominent problem in the public sector. Because of the relative advantages in terms of “job security, power, prestige, career development, fringe benefits, pension, and family-friendly programs” and the significant social prestige and influence, resulting in the “frenzy” for public jobs [6,7,8], public employees who are potentially overqualified (those with Master’s and Doctor’s degrees) for these positions have poured into the public sector [6,9]. For example, 2.5 million individuals applied for the national servant examination in 2023 in China, and the competition ratio for the most competitive position is 6002:1. POQ is prevalent and far-reaching in the public sector [6,10]. Studies have indicated that POQ improves public employees’ work performance and facilitates their realization of ideas [11,12]. However, it can also lead to public employees’ work procrastination, excessive internet usage [10,13], and turnover intention [14]. However, empirical attention to the POQ in public administration is still limited, and more research need to be carried out on the POQ in the public administration field [6].

POQ has both positive and negative effects on the public sector. Because focusing only on one set of responses can lead to a rather fragmentary understanding of the psychology and behavior triggered by the POQ [15], we aim to reconcile the negative and positive consequences of public employees’ POQ from approach (prosocial behavior) and avoidance (deviant behavior) tendencies, reflecting organizational citizenship behavior (OCB) and workplace deviance behavior (WDB), respectively, and building upon the conservation of resource theory (COR). These two behaviors are widely valued and represent typical positive and negative behaviors in the public sector [16,17]. The COR theory posits that individuals tend to conserve, protect, and acquire resources. When individuals experience a loss of resources, they use various strategies to protect themselves and prevent further resource depletion [18]. In the context of POQ, employees may engage in job crafting and seek organizational support to ensure that their qualifications are not wasted [19,20]. On the other hand, people feel a strong sense of unease or concern towards current or potential difficulties, risks, or dangers that occur when individuals experience a loss of their resources. With crisis awareness, people may be prone to negative emotions and behavioral changes, such as depression and job burnout [21]. COR offers a theoretical framework for understanding how POQ elicits behaviors with different action tendencies.

From the approach tendency, it has become a consensus that OCB is one of the most important outcomes of POQ [22]. However, the relationship between POQ and OCB is complicated. Some scholars believe that POQ will reduce OCB, which is mainly based on negative emotions, such as boredom [23,24], negative cognitive perspectives, such as relative deprivation [25], and person-environment fit [26]. Some scholars believe that POQ will promote OCB, which is mainly based on the motivation perspective [27] and positive cognitive perspectives such as self-efficacy, self-expectations [28,29], and relational cognitive [30]. The above discussion rarely considers the individual assessment of the resources. According to this study, individuals with POQ will evaluate their abilities, experience, and other resources, and the results of the evaluation will further cause changes in individual psychology and behavior. The pursuit of control in the work environment is a basic motivation for individuals, and perceived control can be regarded as an individual’s assessment of their physical, social, psychological, and physical resources [31], which is a crucial driver influencing behavior [32]. Therefore, this study introduced perceived control as the mediating variable, arguing that public employees with POQ are prone to producing higher perceived control, which would trigger higher OCB.

From the avoidance tendency, employees with high POQ may display counterproductive behavior [26], which is often used interchangeably with workplace deviance behavior [33]. These studies are mainly based on cognitive feelings (such as relative deprivation) and emotional experiences (such as anger and frustration) [28]. The POQ may have a more profound negative impact on organization through cognition than through emotional changes [34]. Although existing studies focus on cognitive changes such as the sense of fairness and relative deprivation, the above cognition is generated in comparison with others [35]. Not only that, but a common and important central factor in relative deprivation and a sense of equity is the sense of entitlement [35,36]. The authors of this study believe that POQ may affect individuals’ self-cognition and, thus, their work behavior. Meanwhile, this study also extracted the important factor of entitlement and considered psychological entitlement as an explanatory mechanism by which POQ affects WDB.

This research aims to address the aforementioned issues by conducting a time-lagged study with public employees in the Chinese education sector to explore the distinct mechanisms by which prosocial behavior and deviant behavior (organizational citizenship behavior and workplace deviance behavior) may develop as a consequence of POQ. We mainly contribute to the literature in the following three aspects: First, based on the COR theory, we evaluated the role of perceived control and psychological entitlement as unique mediators, linking POQ to approach (OCB) and avoidance (WDB) tendencies, respectively, in response to the academic call for multimediation in POQ research [37,38]. Second, this study provided empirical results for the positive correlation between POQ and OCB from the perspective of the feeling of control, with perceived control as the mediator. Third, from a self-evaluation perspective, this study found that POQ has a negative relationship with workplace deviance behavior through psychological entitlement. Finally, this study focuses on the perspective of POQ in the public sector in China, where it has found fertile breeding ground among the public employees, but still receives less attention from researchers.

### 1.1. Positive Path—Linking POQ to OCB

POQ refers to individuals’ perception that their education, experience, skills, and abilities surpass the requirements of their job, or that their current job lacks opportunities for growth and a platform to showcase their abilities [39], commonly known as being “overqualified and underutilized” or experiencing a “mismatch” between their qualifications and job responsibilities. Extant research has considered positive relationships between POQ and positive behaviors, such as proactive behavior [29].

OCB refers to employees’ voluntary actions that go beyond their formal job responsibilities, with no immediate rewards or incentives, but contribute to organizational performance and efficiency [40]. OCB can be regarded as a socio-political structure and has an exceptionally prominent position in the public sector because the goal of public administration reform is to achieve a greater response of organizations to citizens [41,42]. Existing studies indicate that OCB exists in the public sector, and it is more often for colleagues [43] and more favorable [44]. Operating on a limited budget while maintaining service levels requires effectively stimulating OCB for public employees, which is an important theoretical and practical problem to solve in the complex environment facing the public sector [45].

More and more scholars regard OCB in the public sector as a research topic [42,46,47,48]. Previous studies have widely examined the antecedents of OCB to benefit organizations, mainly including organizational factors and individual factors. Organizational factors include factors such as organizational ethics [49], ethical leadership [50], and organizational dependence [51]. Individual factors include factors such as public service motivation [52] and self-efficacy [53]. In this study, POQ is also an important individual factor in OCB production.

POQ shows an imbalance between resource input and return [54], which means that the employees’ knowledge and skills are not fully utilized, which will waste resources [10]. The COR theory is important for understanding the generation of individual behavior [55]; it posits that individuals employ various strategies to protect themselves from resource loss when facing depletion [18]. The gain paradox principle in this theory also proposes that resource loss will amplify the value of resources, and resource loss has more positive momentum than preventive resources [56]. For example, individuals who experience resource loss will seek appreciation from supervisors or colleagues, leading to OCB [57]. According to the COR theory, individuals with excess qualifications are more likely to perceive the loss of their own resources. At this time, the acquisition of resources is particularly important to them, so they may take some positive actions to supplement the lost resources. Therefore, this study puts forward the following hypothesis:

**H1.** 
*Public employees’ POQ is positively related to OCB.*


Perceived control refers to an employee’s confident judgment of their abilities, believing that they possess sufficient internal resources to influence adverse events and the external environment, and achieve desired outcomes [58]. Perceived control is an individual’s trust in their own abilities; it is a situational assessment and can be used as a resource [31,58]. Job insecurity, self-affirmation, loneliness, and social support affect perceived control [31,59,60,61].

According to the COR theory, people attempt to preserve, develop, and safeguard their resources, which include all objects, conditions, personal traits, and energy necessary for individual survival. This theory also points out that the situation of resource loss magnifies the value of resources [18,62]. Public employees with POQ are very confident in their abilities and will generate a sense of waste of resources when their abilities are not well used [10,63]. They will be more sensitive to the resources they can have and value the resources they own, such as their abilities and experiences. Their appraisal of resources will improve with emphasis and confidence, giving them greater perceived control. Hence, public employees with POQ will have a stronger sense of control. Therefore, this study puts forward the following hypothesis:

**H2.** 
*Public employees’ POQ is positively related to perceived control.*


The COR theory suggests that when individuals have ample resources, they are motivated to cultivate and enhance the value of those resources [62]. For example, individuals with high psychological capital are willing to invest resources into innovative behaviors that can be reported to increase the stock of resources [64]. A sense of control makes individuals more confident and optimistic that they will experience positive events in the future [65,66], and they are willing to invest in their future [67]. Hence, when perceived control is high, individuals may invest their excess psychological resources in positive behaviors, creating a spiral effect of resource gain. Previous studies on perceived control and positive behavior have also provided side support. For example, a meta-analysis by Rudolph et al. (2004) found a positive correlation between perceived control and prosocial behavior [68]. Based on the above analysis, the following hypothesis is proposed:

**H3.** 
*Public employees’ perceived control is positively related to OCB.*


Combining H2 and H3, this study argues that perceived control plays a mediating role in the potential impact of public employees’ POQ on OCB. When public employees perceive overqualification in the organization, they believe that their skills, knowledge, and other resources are wasted. They cherish the existing resources, like the perceived control, and they hope to invest their resources in engaging in OCB to obtain resource gains. In summary, the following hypothesis is proposed:

**H4.** 
*Perceived control mediates the relationship between POQ and OCB.*


### 1.2. Negative Path—Linking POQ to WDB

WDB refers to intentional behaviors of employees in the workplace that cause harm to the organization or others [69,70], adversely affecting the organization’s interests [71]. Previous research has identified various factors that influence employees’ willingness to engage in WDB, such as cognitive appraisal, affective response, negative affectivity, job pressure, and authoritarian leadership [72,73,74]. In this study, the POQ is also an important factor affecting the production of WDB among public employees.

The COR theory posits that individuals tend to protect their resources. When resources are lost, individuals can experience negative emotions such as nervousness and anxiety. Mainly when resources are depleted, individuals may engage in irrational behaviors as a means of self-defense [18,62]. Employees with POQ are susceptible to experiencing emotions such as dissatisfaction due to unmet expectations [21], which can lead to the loss of their resources. In such situations, individuals may engage in avoidance behaviors, absenteeism, or other behaviors not conducive to organizational development to compensate for unfulfilled expectations [75]. Existing research has suggested that individuals who feel overqualified exhibit pro-organizational unethical behaviors or deviant innovation behaviors that deviate from ethical norms [10,21]. Therefore, for public employees with POQ, resource loss induced by a mismatched job may prompt them to engage in deviant behaviors in the workplace. Based on this, the following hypothesis is proposed:

**H5.** 
*Public employees’ POQ positively impacts public employees’ WDB.*


We propose that psychological entitlement plays a mediating role in the potential impact of public employees’ POQ on WDB. Psychological entitlement refers to a stable personality trait in individuals where they believe they should receive more and lose less [76,77,78], but it is a fluctuating state [79]. Some scholars also believe that psychological privilege is a stable and universal subjective perception that individuals feel entitled to preferential treatment [80]. Collectively, its stable state is seen as an aspect of personality, while its change represents the cognitive side. Psychological entitlement is based on many favorable self-perceptions and optimistic expectations [81], such as inflated self-perceptions, over-identification, and heightened expectations [82,83,84].

Public employees with POQ will feel resource loss because the organizational environment limits their abilities [10,64]. According to the COR theory, people desire to obtain something significant to them, especially when resources are depleted [56]. Hence, public employees with POQ prefer access to the resources. Psychological entitlement often leads people to think they should have more resources than others [85]. On the one hand, public employees with POQ feel confident in their abilities and develop superiority [29], resulting in higher self-evaluation and further psychological entitlement [82]. On the other hand, public employees with POQ may feel unfairness [24], which can also contribute to higher psychological entitlement among public employees [86]. Based on this, the following hypothesis is proposed:

**H6.** 
*Public employees’ POQ positively impacts psychological entitlement.*


Psychological entitlement, as a cognitive bias, has been found to have negative impacts, such as leading to more knowledge-hiding behavior, political behavior, and co-worker abuse [87,88,89]. According to the COR theory, in the case of cognitive dissonance, individuals will require more resources to control their emotions, resulting in resource imbalance [90] and causing the adjustment of attitudes, cognition, and behavior; especially in the situation of resource loss, individuals may adopt irrational behaviors [56]. Public employees with high psychological entitlement are more likely to be self-centered, demanding more than they give, distrustful of others, greedy, selfish, and lacking empathy [91]. They believe they deserve preferential treatment regardless of performance [76,92,93]. When public employees’ inflated psychological needs are unmet, they may engage in negative behaviors to offset unmet expectations [76,94]. Based on this, the following hypothesis is proposed:

**H7.** 
*Public employees’ psychological entitlement positively correlates with WDB.*


Combining H6 and H7, this study suggests that psychological entitlement plays a mediating role between POQ and workplace deviance behavior. When individuals have POQ within the organization, the perceived gap in treatment and status may induce psychological entitlement [29] and trigger feelings of anger [95], frustration, and loss [87], resulting in a depletion of psychological resources and the manifestation of negative behaviors. Based on the literature, the following hypothesis is proposed:

**H8.** 
*Psychological entitlement mediates the relationship between POQ and WDB.*


The proposed model is depicted in Figure 1.

## 2. Materials and Methods

### 2.1. Participants

Intermediation analysis based on cross-sectional data is not suitable for causal inference and requires longitudinal tracking [96]. Referring to the practices in the multistage study [94], the data were collected through a multi-point (2-week) questionnaire survey, and samples were obtained from public employees in the education field (teachers with official staffing recommendations) in three schools in two counties in northern China. Participants were invited to fill in demographic information and the POQ scale at point 1 (T1), the scales of perceived control and psychological entitlement at point 2 (T2), and the scales of organizational citizenship behavior and workplace deviance behavior at point 3 (T3). Each measurement was separated by two weeks and lasted for three days. A total of 500 public employees were invited to participate in this study, and questionnaires were distributed in three rounds. In the first round, 428 questionnaires were gathered, with an effective response rate of 85.60%. In the second round, 500 questionnaires were distributed, and 426 valid questionnaires were collected, with an effective response rate of 85.20%. In the third round, 500 questionnaires were distributed, and 421 valid questionnaires were collected, with an effective response rate of 84.20%. The sample’s demographic characteristics are as follows: male: 27.8%, female: 72.7%; the mean age was 38 years old (standard deviation of 0.96); a bachelor’s degree or above is mainly required (96.9%).

### 2.2. Measurements

The measurement instruments in this study are widely used in the existing literature. Following the translation and back-translation process, the Chinese versions of the items were prepared, and a 5-point Likert scale was used for evaluation (1 = “strongly disagree”, 5 = “strongly agree”).

#### 2.2.1. Perceived Overqualification

Perceived overqualification was measured using the 4-item scale developed by Johnson, G., and Johnson, W. (1996) [97]. A representative item is as follows: “I am overqualified for the job I hold”. The Cronbach’s α of the scale in this study was 0.82.

#### 2.2.2. Perceived Control

Perceived control was measured using the 3-item scale developed by Susan et al. (1989) [98]. A representative item is as follows: “I have enough power in this organization to control events that might affect my job”. The Cronbach’s α of the scale in this study was 0.84.

#### 2.2.3. Psychological Entitlement

Psychological entitlement was measured using the 4-item scale developed by Yam et al. (2017) [99]. A representative item is as follows: “I honestly feel I’m just more deserving than others”. The Cronbach’s α of the scale in this study was 0.93.

#### 2.2.4. Organizational Citizenship Behavior

Organizational citizenship behavior was measured using the 8-item scale developed by Lee and Allen (2002) [100]. A representative item is as follows: “Attend functions that are not required, but that help the organizational image”. The Cronbach’s α of the scale in this study was 0.92.

#### 2.2.5. Workplace Deviance Behavior

Workplace deviance behavior was measured using the 10-item scale developed by Qin et al. (2020) [94]. A representative item is as follows: “Complained about insignificant things at work”. The Cronbach’s α of the scale in this study was 0.98.

#### 2.2.6. Demographics

The study also included gender, age, and education level as control variables. Previous studies have reported that these demographic variables were significantly correlated with the outcome variables, organizational citizenship behavior, and workplace deviance behavior [101,102].

## 3. Results

### 3.1. Common Method Bias

Harman’s single-factor test was conducted to test whether this study had a common method bias. An exploratory factor analysis of five variables’ components using SPSS23.0 software generated a total of five factors with eigenvalues larger than one. The most significant component explained 31.66% of the variation. The majority of the variation could not be explained by a single cause.

This study also used the unmeasured latent method construct (ULMC) to test the common method bias [103]. After adding a latent construct to the five-factor model, the χ^2^/*df*, comparative fit index (CFI), Tucker–Lewis index (TLI), and root-mean-square error of approximation (RMSEA) of the ULMC model and the five-factor model varied and did not exceed 0.05 (Table 1). As a result, there was no common method bias concern in our study [104].

### 3.2. Reliability and Construct Validity

Confirmatory factor analyses were conducted using the statistical program Mplus 8.3 to compare the fit indices of the five-factor model with those of other combination models in order to test the discriminant validity of the variables [105]. The results (in Table 1) revealed that the five-factor model fit best (χ^2^/*df* = 3.55, CFI (comparative fit index) = 0.93, TLI (Tucker–Lewis index) = 0.93, RMSEA (root-mean-square error of approximation) = 0.08, SRMR (standardized root mean square residual) = 0.05). In contrast, the one-factor model fitted the poorest (χ^2^/*df* = 33.57, CFI = 0.06, TLI = 0.05, RMSEA = 0.28, SRMR = 0.34).

Table 2 shows the reliability and validity results. The Cronbach’s alpha, CR, and AVE values for the measuring scales of POQ, perceived control, psychological entitlement, OCB, and WDB are above the thresholds of 0.7, 0.7, and 0.5, respectively, indicating strong internal consistency and reliability. In addition, the square roots of AVE for the correlation coefficients across variables are stronger for all variables, confirming the scales’ excellent discriminant validity.

### 3.3. Descriptive Statistics and Correlational Analysis

Table 3 shows the important variables’ means, standard deviations, and Pearson correlation coefficients. POQ was positively correlated with perceived control (*r* = 0.17, *p* < 0.001), positively correlated with psychological entitlement (*r* = 0.20, *p* < 0.001), and positively correlated with OCB (*r* = 0.15, *p* < 0.01), while POQ was not significantly correlated with workplace deviance behavior (*r* = 0.08, *p >* 0.05). Perceived control was positively correlated with OCB (*r* = 0.30, *p* < 0.001). Psychological entitlement was positively correlated with workplace deviant behavior (*r* = 0.22, *p* < 0.001). According to Cohen’s (1988) suggestion, our results are almost consistent with our theoretical expectations [106].

### 3.4. Hypothesis Testing

Hierarchical regression was first used to test the two intermediaries separately. The results are presented in Table 4. After controlling for the three demographic variables of gender, age, and education level (M1), the main effects among the variables were tested. The results showed that POQ had a significantly positive impact on perceived control (M2: *β* = 0.15, *p* < 0.01) and on OCB (M4: *β* = 0.13, *p* < 0.01). Perceived control also positively affected OCB (M5: *β* = 0.29, *p* < 0.001). Thus, H1, H2, and H3 are supported. After setting perceived control as an independent variable, the regression coefficient for POQ was no longer significant (M6: *β* = 0.09, *p* > 0.05), whereas perceived control continued to have a significant positive influence (M6: *β* = 0.27, *p* < 0.001). The fact that perceived control is a full mediator of POQ influencing OCB is shown by the ∆R^2^ value of 0.07. H4 is therefore supported. Psychological entitlement had no significant impact on OCB (M7: *β* = −0.07, *p* > 0.05).

After controlling for the three demographic variables of gender, age, and education level (M9), the main effects among the variables were tested. The results showed that POQ had a significantly positive impact on psychological entitlement (M10: *β* = 0.19, *p* < 0.001), but it had no significant impact on WBD (M12: *β* = 0.06, *p* > 0.05). Psychological entitlement positively affected WBD (M13: *β* = 0.20, *p* < 0.001). After setting psychological entitlement as an independent variable, the regression coefficient for POQ was no longer significant (M14: *β* = 0.03, *p* < 0.60), whereas psychological entitlement continued to have a significant positive influence (M14: *β* = 0.20, *p* < 0.001). Thus, H6 and H7 are supported, while H5 and H8 are not supported. Perceived control had no significant impact on WBD (M15: *β* = 0.0, *p* > 0.05).

Because this is a two-pathway model, utilizing hierarchical regression alone does not adequately evaluate the effect when two lines coexist, so we built structural equation models for further hypothesis testing.

To verify the multiple mediation model, we further constructed the structural equation model in Mplus 8.3 with 5000 repeated samplings with returns (bootstrapping method). The results are presented in Table 5. The effect of POQ on OCB was not significant (*b* = 0.09, *p* > 0.05), not supporting H1, while POQ had a significantly positive impact on perceived control (*b* = 0.16, *p* < 0.01), supporting H2. Perceived control had a significantly positive impact on OCB (*b* = 0.25, *p* < 0.001), supporting H3. As shown in Table 6, the indirect effect of the path (POQ-PC-OCB) was 0.04 (*p* < 0.05), indicating that perceived control played a fully mediating role in the path, and H4 was confirmed. Psychological entitlement had a significantly positive impact on OCB (*b* = −0.14, *p* < 0.05), and we further validated its mediating effect between POQ and OCB: the indirect effect (POQ-PE-OCB) was −0.03 (*p* > 0.05), indicating that psychological entitlement played no mediating role in the path.

The effect of POQ on WBD was not significant (*b* = 0.03, *p* > 0.05), not supporting H5, while POQ had a significantly positive impact on psychological entitlement (*b* = 0.21, *p* < 0.01), supporting H6. Psychological entitlement had a significantly positive impact on WBD (*b* = 0.22, *p* < 0.01), supporting H7. As shown in Table 6, the indirect effect of the path (POQ-PE-WBD) was 0.04 (*p* < 0.05), indicating that psychological entitlement played a fully mediating role in the path, and H8 was confirmed.

## 4. Discussion

POQ has become a common workplace phenomenon. On the one hand, economic downturns have led to an increasing number of higher education graduates or experienced job seekers who are forced to work outside the field they studied or in less desirable employment [107]. On the other hand, research has revealed the negative effects of POQ on the focal employees, such as turnover intention [108], and lower job satisfaction [2,9,109,110]. However, due to the general improvement of education level, the reality of difficult job hunting, and the positive influence of POQ, such as the fact that individuals with high POQ may complete their work faster and more effectively, leading to increased self-efficacy and performance [111], employers are also willing to offer the same opportunities and hire job seekers with higher ability and experience than the job requirements [112]. This inevitable phenomenon makes the impact of POQ on employee psychology and behavior increasingly important [113].

Although the influence of POQ is widely recognized, current research still has some shortcomings. First, previous studies have focused on POQ in the private sector but ignored the public sector, where POQ may be more common, especially in countries with highly respected public professions like China. Secondly, previous studies have extensively explored the negative and positive effects of POQ on employee psychology and behavior, but there is a lack of integrated research on the double-edged sword effect of POQ. Finally, considering the definition of POQ, namely, that employees think that their ability and level are higher than the requirement of post [1,2], when the individuals produce a higher POQ, their assessment of the resources will improve, resulting in a higher sense of control. They are also keen to get resources and produce psychological entitlement by comparing themselves, which has been ignored in previous studies.

Hence, focusing on these research gaps, in this study, by introducing the concepts of perceived control and psychological entitlement, we constructed a double-edged framework to understand the cognitive and behavioral outcomes of public employees. Based on the COR theory, we suggested that public employees’ POQ influences their OCB through perceived control; not only that, but public employees’ OCB will also affect their WDB through psychological entitlement. The empirical results of a time-lagged study with 421 public employees provided support for this theoretical framework.

### 4.1. Theoretical Implications

These findings provide significant theoretical advances in the literature on overqualification. First, in a research framework that incorporates two mediating mechanisms, this study investigates the positive influence of POQ on OCB and the negative effect of POQ on WDB. This work responds not only to Erdogan and Bauer’s (2021) call for multi-mediating research but also to Li et al.’s (2021) call for integrating the double-edged sword mechanism [22,28]. Because of the prestige and welfare benefits provided to employees of Chinese public organizations, as well as the fierce rivalry for employment opportunities in public organizations, POQ may emerge in the public sector [6]. While previous research has looked at both the positive and negative effects of POQ [9,19,20], few studies have specifically looked at the public sector from both the positive and negative effects of POQ within a single research framework [28]. This study has great theoretical significance for understanding the impact of public employees’ POQ on their workplace behavior and its underlying mechanisms because it considers both the positive and negative effects of POQ in a resource conservation framework. Subsequently, further research on POQ in the public sector is needed to explore which cognitions, emotions, or motivations are more influenced by POQ, thus having a stronger effect on the positive or negative behaviors of public employees. Furthermore, influenced by cultural factors, the degree of competition for public sector employment opportunities is different, as is POQ. For example, in Confucius cultures, such as China and Korea, public employment opportunities are more competitive [6]. Therefore, future studies on POQ in the public sector can also consider the influence of cultural factors.

Second, the approach results showed that while POQ was unable to produce a direct correlation with OCB, it might nevertheless support OCB by increasing the perceived control of public employees. The findings about the relationship between POQ and OCB are contradictory; some research [23,114] indicates a negative correlation between the two, while other studies [27,30] indicate a positive correlation. We believe that different mechanisms of impact are the cause of the conflicting conclusions, based on the results of this study and the research that has already been carried out. For instance, while encouraging a positive incentive to attain status, POQ raises OCB [27]. Comparatively speaking, nevertheless, POQ is probably going to result in negative cognitive bias, which lowers OCB [30]. This is also supported by the data results from the available studies [27,30]. It can be seen that the positive or negative mechanism between POQ and OCB requires further discussion among scholars. While this study provides empirical evidence for POQ’s role in promoting OCB in the public sector from the perspective of the sense of control generated by resource evaluation, in order to determine when and under what circumstances POQ will have a negative or positive impact on OCB, future research can examine the two distinct effect boundary conditions, contrast POQ’s negative and positive effect mechanisms and strengths, and further develop POQ’s influence on OCB parallel mediation.

Finally, the avoidance results showed that while POQ was unable to produce a direct correlation with WDB, it might nevertheless support WDB by increasing the psychological entitlement of public employees. According to Maynard et al. (2015) [115], narcissism—particularly the entitlement dimension—plays a significant role in the development of POQ. According to POQ’s influence results, people’s relative experience of deprivation may be impacted, and entitlement plays a significant role in this notion [35]. Individuals with POQ believe that their abilities exceed job requirements, which may lead them to expect higher performance pay or status. When these expectations are not met, individuals may engage in irrational behaviors. Psychological entitlement is also a reaction of self-evaluation. Existing studies have indicated that POQ can trigger psychological entitlement and subsequently lead to pro-organizational unethical behaviors [116]. According to this study, psychological entitlement is also an important psychological factor that leads to the negative behavior of public employees with POQ. This study also responds to calls for more consideration of psychological entitlement in organizations [93]. In order to prevent reverse causality, this study used longitudinal tracking. However, more research is needed to fully explore the relationship between psychological privilege and POQ, as well as to clarify the causal relationship itself. In addition, it is well established that psychological entitlement has detrimental effects [117,118]. It is also possible to further discuss the conditions under which psychological entitlement can be lessened, as well as the conditions under which psychological entitlement’s detrimental effects can be mitigated. The relationship between psychological entitlement and good conduct can also be investigated, as research has shown that psychological entitlement might increase job involvement by raising career ambitions [119].

### 4.2. Practical Implications

The present research has several implications for practitioners. By examining the constructive and destructive behavioral outcomes of overqualification, this study emphasizes the complexity of overqualification. Especially considering that overqualification has become more common in recent years [6], managers must understand and adequately manage overqualification to take advantage of its beneficial implications and mitigate its detrimental effects on public employees. Our results show that POQ can lead to different workplace behaviors by causing different psychological cognitions among individuals. It is therefore necessary to differentiate between individuals producing POQ.

Firstly, overqualified public employees perceived themselves as having more job control, leading to more OCB. In recent years, OCB has been encouraged in the public sector [17]. According to our results, public employees with high POQ should be responsible for more important tasks and given more opportunities for visits, study, exchanges, and secondments to meet their growth needs, and to avoid making them think that they are snubbed. Through guidance and support, managers can help overqualified public employees satisfy their perceived control, enhancing their sense of belonging to the organization.

Secondly, increased WDB induced by the psychological entitlement of overqualified public employees indicated that POQ could not directly lead to WDB, especially in the absence of psychological entitlement. Managers should carefully engage in timely communication with qualified public employees with an increasingly higher level of psychological entitlement. WDB will result in many losses in resources and productivity for organizations [120], which needs to be brought to organizational attention. If overqualified public employees have already been employed, our findings highlight the need to consider placing them in positions where they are less likely to feel overqualified. Multiple measures, such as communication, education, and training, can be implemented to guide them away from negative behaviors and prevent “good employees” from engaging in actions that are detrimental to the organization.

Lastly, to solve the problem of POQ, in addition to the efforts of organizations, public employees also need to adjust their own cognitions and behaviors. Public employees themselves could do the job crafting. They should choose positions that align with their educational background, skills, experience, interests, needs, and development plans. They can use their spare time and energy to innovate in their work and strive for rights and privileges that set them apart from others. They can search for opportunities for further education and training, exchanges and secondments, and job title promotions, while their employers can leverage compliance-based rights as a foundation to optimize their public employees within the organization.

### 4.3. Research Limitations and Future Directions

The findings and conclusions of this study should be interpreted in light of several limitations. Firstly, this study explores the mechanism through which POQ affects workplace behavior without considering the boundary conditions of this effect. Future research could incorporate organizational situational factors or individual characteristics as potential boundary conditions. For example, individuals with high organizational identification may exhibit a stronger sense of belonging to the organization and act in accordance with its goals [121]. Therefore, even with POQ, they may willingly take on responsibilities and obligations [122]. Psychological empowerment is a psychological resource that is produced as a result of numerous work value signals provided by the organization, which affect employees’ attitudes and behaviors [123]. Employees with high psychological empowerment possess more psychological resources, such as work security, confidence, and basic psychological needs, which will affect their resource input behaviors [14,62]. As a complementary resource, psychological resources can assist employees in demonstrating greater resilience in the face of resource loss, thus buffering the many negative consequences [14,56].

Secondly, the research methods employed in this study may be insufficient. Although data were collected at three time-points, all variables in our model were self-reported, which may introduce common method bias to some extent. We believe that self-reported measures are appropriate for our research question because individuals have the best knowledge about their abilities, cognitions, and behaviors, and thus, self-reported measures provide a valid approach to capture the information we are interested in [124,125]. We also utilized several strategies to mitigate common method variance and self-reporting bias, including a time-lagged design, highlighting to participants that data were only used for research purposes, and ensuring the participants’ confidentiality [126]. Future research is needed to replicate our results using objective indicators of overqualification and enhance data reliability by incorporating data from both leaders and subordinates. In addition, the sampling of research subjects may be limited. Due to economic factors and regional development, POQ among public employees may vary in different regions. Future research could consider expanding the geographical scope, such as considering regional economic development as a factor.

## 5. Conclusions

The current study makes an effort to look at how perceived overqualification among public employees can generate cognitive and behavioral outcomes. This study revealed that POQ can boost OCB via perceived control (positive) and WDB via psychological entitlement (negative). Our findings make significant contributions to overqualification management research and practice in the public sector.

## Figures and Tables

**Figure 1 behavsci-14-00048-f001:**
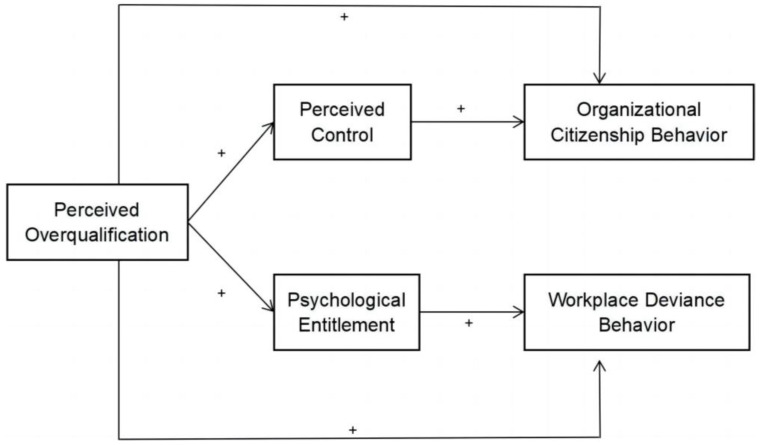
Research model. Note. + represents a positive correlation.

**Table 1 behavsci-14-00048-t001:** Confirmatory factor analysis results of the competition model (*N* = 421).

Model	Factor Combination	χ^2^	*df*	χ^2^/*df*	CFI	TLI	RMSEA	SRMR
ULMC model	POQ, PC, PE, OCB, WDB, latent construct	1201.73	336	3.58	0.94	0.93	0.08	0.05
Five-factor	POQ, PC, PE, OCB, WDB	1287.61	363	3.55	0.93	0.93	0.08	0.05
Four-factor	POQ, PC + PE, OCB, WDB	3068.96	371	8.27	0.81	0.78	0.13	0.12
Three-factor	POQ, PC, PE + OCB + WDB	5176.58	374	13.84	0.65	0.62	0.18	0.19
Two-factor	POQ, PC + PE + OCB + WDB	5802.87	376	15.43	0.61	0.58	0.19	0.19
One-factor	POQ, PC, PE, OCB, WDB	13,493.36	402	33.57	0.06	0.05	0.28	0.34

Note. + represents the two factors merging into one. “POQ” = “perceived overqualification”; “PC” = “perceived control”; “PE” = “psychological entitlement”; “OCB” = “organizational citizenship behavior”; “WDB” = “workplace deviance behavior”.

**Table 2 behavsci-14-00048-t002:** Reliability and validity analysis (*N* = 421).

Constructs	Items	Factor Loadings	Cronbach’s α	CR	AVE
POQ	My formal education over qualifies me for my present job	0.64	0.82	0.88	0.66
	My talents are not fully utilized on my job	0.76			
	My work experience is more than necessary to do my present job	0.90			
	Based on my skills, I am overqualified for the job I hold	0.91			
PC	I have enough power in this organization to control events that might affect my job	0.81	0.84	0.91	0.77
	In this organization, I can prevent negative things from affecting my work situation	0.90			
	I understand this organization well enough to be able to control things that affect me	0.92			
PE	I honestly feel I’m just more deserving than others	0.91	0.93	0.95	0.82
	Great things should come to me	0.93			
	I demand the best because I’m worth it	0.88			
	I deserve more things in my life	0.92			
OCB	Attend functions that are not required but that help the organizational image	0.69	0.92	0.94	0.65
	Keep up with developments in the organization	0.83			
	Defend the organization when other employees criticize it	0.83			
	Show pride when representing the organization in public	0.85			
	Offer ideas to improve the functioning of the organization	0.68			
	Express loyalty toward the organization	0.79			
	Take action to protect the organization from potential problems	0.87			
	Demonstrate concern about the image of the organization	0.87			
WDB	Purposely wasted the employer’s materials/supplies	0.93	0.98	0.98	0.87
	Complained about insignificant things at work	0.82			
	Told people outside the job what a lousy place you work for	0.90			
	Came to work late without permission	0.90			
	Stayed home from work and said you were sick when you weren’t	0.96			
	Insulted someone about their job performance	0.97			
	Made fun of someone’s personal life	0.97			
	Ignored someone at work	0.96			
	Started an argument with someone at work	0.94			
	Insulted or made fun of someone at work	0.96			

Note. “POQ” = “perceived overqualification”; “PC” = “perceived control”; “PE” = “psychological entitlement”; “OCB” = “organizational citizenship behavior”; “WDB” = “workplace deviance behavior”.

**Table 3 behavsci-14-00048-t003:** Means and standard deviations (SD) (*N* = 421).

Variables	Mean	SD	1	2	3	4	5	6	7	8
1 Gender_T1	1.72	0.45								
2 Age_T1	3.35	0.96	−0.11 *							
3 Education level_T1	5.10	0.49	0.15 **	−0.13 **						
4 POQ_T1	3.04	0.66	−0.17 ***	0.16 **	−0.07	(0.44)				
5 PC_T2	3.24	0.71	−0.10 *	0.12 *	−0.06	0.17 ***	(0.59)			
6 PE_T2	2.72	0.74	−0.08	0.02	−0.00	0.20 ***	0.16 **	(0.67)		
7 OCB_T3	3.46	0.66	−0.04	0.16 **	−0.04	0.15 **	0.30 ***	−0.07	(0.42)	
8 WDB_T3	1.47	0.82	−0.22 ***	−0.07	0.07	0.08	0.02	0.22 ***	−0.12 *	(0.76)

Note. *, *p* < 0.05, **, *p* < 0.01, ***, *p* < 0.001. Gender: 1 male, 2 female, Age: 1 = 20 years old and below, 2 = 21–30 years old, 3 = 31–40 years old, 4 = 41–50 years old, 5 = 51–60 years old, 6 = 60 years old or above. Education level: 1 = primary school and below, 2 = junior high school, 3 = high school (technical secondary school, high vocational skills), 4 = junior college, 5 = undergraduate college, 6 = postgraduate, 7 = doctor or above. “POQ” = “perceived overqualification”; “PC” = “perceived control”; “PE” = “psychological entitlement”; “OCB” = “organizational citizenship behavior”; “WDB” = “workplace deviance behavior”. The square root of AVE is on the diagonal.

**Table 4 behavsci-14-00048-t004:** Regression analysis (*N* = 421).

	Variables	Perceived Control			Organizational Citizenship Behavior					Psychological Entitlement			Workplace Deviance Behavior				
		M1	M2	M3	M4	M5	M6	M7	M8	M9	M10	M11	M12	M13	M14	M15	M16
Control Variables	Gender	−0.09	−0.06	−0.02	0.00	0.01	0.02	−0.02	−0.002	−0.08	−0.06	−0.24 ***	−0.23	−0.22 ***	−0.22 ***	−0.24 ***	−0.23 ***
	Age	0.11 **	0.09	0.16 **	0.14 **	0.13 **	0.11 **	0.16 **	0.14 *	0.01	−0.02	−0.08	−0.09	−0.08	−0.09	−0.08	−0.09
	Education level	−0.03	−0.03	−0.01	−0.01	−0.01	0.00	−0.01	−0.009	0.01	0.01	0.10	0.10 *	0.09 *	0.09 *	0.10 *	0.10 *
Independent Variables	Perceived Overqualification	—	0.15 **	—	0.13 **	—	0.09	—	0.15 **	—	0.19 ***	—	0.06	—	0.03		0.06
Mediator	Perceived Control	—	—	—	—	0.29 ***	0.27 ***	—	—	—	—	—	—	—	—	0.01	0.001
	Psychological Entitlement	—	—	—	—	—	—	−0.07	−0.10 *	—	—	—	—	0.20 ***	0.20 ***		
	R^2^	0.02	0.04	0.03	0.04	0.11	0.11	0.03	0.05	0.01	0.04	0.06	0.07	0.10	0.11	0.06	0.07
	△R^2^	0.02	0.02	0.03	0.02	0.08	0.07	0.01	0.01	0.01	0.03	0.06	0.004	0.04	0.04	0.00	0.004
	F	3.31 **	4.75 ***	3.72 *	4.64 ***	12.23 ***	10.69 ***	3.31 *	4.52 **	1.00	4.47 **	9.47 ***	7.53 ***	12.09 ***	9.71 ***	7.10 ***	6.01 ***

Note. *, *p* < 0.05, **, *p* < 0.01, ***, *p* < 0.001.

**Table 5 behavsci-14-00048-t005:** Results of the mediating effects (*N* = 421).

	Perceived Control		Organizational Citizenship Behavior		Psychological Entitlement		Workplace Deviance Behavior	
	*b*	*p*	*b*	*p*	*b*	*p*	*b*	*p*
Gender	−0.06	0.003 **	0.02	0.68	0.19	0.001 ***	−0.22	0.000 ***
Age	0.09	0.173	0.11	0.02	−0.06	0.051	−0.09	0.053
Education Level	−0.03	0.090	−0.003	0.93	−0.02	0.693	0.09	0.047 *
Perceived Overqualification	0.16	0.004 **	0.09	0.085	0.21	0.001 ***	0.03	0.651
Perceived Control			0.25	0.000 ***			−0.03	0.644
Psychological Entitlement			−0.13	0.014 *			0.22	0.001 ***

Note. *, *p* < 0.05, **, *p* < 0.01, ***, *p* < 0.001.

**Table 6 behavsci-14-00048-t006:** Mediator path analysis (*N* = 421).

Path		Effect	SE	*p*
POQ→PC→OCB	Total effect (c1)	0.13	0.06	0.02
	Direct effect (c1′)	0.09	0.05	0.09
	Indirect effect (a1b1)	0.04	0.02	0.02
POQ→PE→WBD	Total effect (c2)	0.06	0.06	0.29
	Direct effect (c2′)	0.03	0.06	0.65
	Indirect effect (a2b2)	0.04	0.02	0.02

Note. “POQ” = “perceived overqualification”; “PC” = “perceived control”; “PE” = “psychological entitlement”; “OCB” = “organizational citizenship behavior”; “WDB” = “workplace deviance behavior”. “c” represents the sum of direct and indirect effects; “c’” represents the direct effect of the independent variable on the dependent variable; “ab” represents the indirect effect of the independent variable on the dependent variable via the mediating variable.

## Data Availability

All data included in the current study can be obtained from the corresponding author upon reasonable request.

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
