# Peer review of "Does Overqualification Play a Promoting or a Hindering Role? The Impact of Public Employees’ Perceived Overqualification on Workplace Behaviors"

_behavsci, 2024, doi:10.3390/bs14010048_

Round 1

Reviewer 1 Report

Comments and Suggestions for Authors

Overall, this manuscript is interesting. The manuscript can stand the test of time. however, there are a few concerns that should be addressed:

1. The introduction should establish the research gaps and discuss why addressing them is important.

2. The literature is built on relevant literature. However, I suggest the authors discuss the hypotheses separately rather than all of them together. Provide literature support for each hypothesis proposed.

3. The methodology should be further explained. What was the duration of each phase? How many were collected in each phase? How were the surveys distributed? The rigor of the methodology should be improved by providing detailed information for other researchers to replicate.

4. Results are explained well. However, using Harman's single-factor test to assess method bias is insufficient.

5. Discussions are limited. The findings should be discussed in light of existing literature and theory. what is new and interesting that makes this manuscript unique should be discussed.

6. Theoretical contributions and practical implications are limited in their current form.

Comments on the Quality of English Language

Satisfactory.

Reviewer 2 Report

Comments and Suggestions for Authors

Dear  authors, 

I perceive that this paper is still in the preliminary stage and is not yet prepared for publication due to the following reasons.

1. Your headline caused confusion among readers from the outset. Are you referring to the concept of "Dose Overqualification" or "Does Overqualification"?

2. The abstract contains the outcomes of statistical computations. If the significance is there, what is the consequence or importance of it? You need to provide more detailed explanations instead of just stating whether something has significance or not. 

3. The previous literary works failed to adequately align with the contemporary trends in Public Administration reform, including the principles of New Public Management. 

4. OCB has garnered significant attention in public sectors. You attempt to establish a connection between overqualification and organisational citizenship behaviour (OCB) by hypothesising that individuals who view themselves as overqualified will exhibit higher levels of OCB. Your paper lacks a coherent rationale and solid basis for explaining why public sector personnel would exhibit such behaviour.  

5. The connection between the manuscript's goal and the justification for the research was inadequately established. The introduction should possess a cohesive style and provide a clear explanation of the topic at hand, as well as a comprehensive overview of the existing academic literature in this field and the specific gap that currently exists. Furthermore, the present study  does not addresses the existing research gap, only collection of theories.

6. In order to substantiate your claim about doing a confirmatory factor analysis (CFA), it is necessary to provide indicators that can determine which items correspond to the five factors you mentioned (POQ, PC, PE, OCB, WDB). However, no such indicators were supplied.   

7. The conclusion is overly simplified, failing to address the research objective and merely asserting the contribution without providing any future study directions or limitations.

Reviewer 3 Report

Comments and Suggestions for Authors

Summary: The paper investigates a link between workers’ perceived over-qualification and a) subjective cognition (perceived control and entitlement) and b) behavior (approach and avoidance tendencies) using data on 421 public employees.  Empirical analysis suggests positive relationships for the links mentioned above.  The topic is relevant in that it explores the channels through which perceived over-qualification might affect labor market outcomes like wages, productivity, absenteeism, and employment. 

Comments

1.      Prevalence of over-qualification would seem to suggest that employers are getting it wrong when setting hiring standards.  Can the author mention and discuss this issue in the paper?

2.      On page 5, in section 2.1, the authors describe the gap of two weeks as data were being collected. What is importance of the two –week gap?

3.      Does "education field" mean that the workers in the sample are teachers? Can you give us a better sense as to the type of work the survey respondents are doing?

4.      Add a control to differentiate between workers samples from the two different cities.

Round 2

Reviewer 2 Report

Comments and Suggestions for Authors

Dear  authors, 

The article has been   improved considerably. However, I still require   that  you define   what are  items   in each  Construct. For example  you just wrote POQ1 , POQ2,   POQ3  and POQ4 for Perceived Overqualification. But  what are they? The same things  should be applied   to the rest of constructs.
